# WavReward: Spoken Dialogue Models With Generalist Reward Evaluators

## Abstract

End-to-end spoken dialogue models such as GPT-4o-audio have recently garnered significant attention in the speech domain. However, the evaluation of spoken dialogue models' conversational performance has largely been overlooked. This is primarily due to the intelligent chatbots convey a wealth of non-textual information which cannot be easily measured using text-based language models like ChatGPT. To address this gap, we propose **WavReward**, a reward feedback model based on audio language models that can **evaluate both the IQ and EQ of spoken dialogue systems with speech input**. Specifically, 1) based on audio language models, WavReward incorporates **the deep reasoning process and the nonlinear reward mechanism** for post-training. By **utilizing multi-sample feedback via the reinforcement learning** algorithm, we construct a specialized evaluator tailored to spoken dialogue models. 2) We introduce ChatReward-30K, a preference dataset used to train WavReward. ChatReward-30K includes both comprehension and generation aspects of spoken dialogue models. These scenarios span various tasks, such as text-based chats, nine acoustic attributes of instruction chats, and implicit chats. WavReward outperforms previous state-of-the-art evaluation models across multiple spoken dialogue scenarios, achieving a substantial improvement about Qwen2.5-Omni in objective accuracy from 53.4% to 91.5%. In subjective A/B testing, WavReward also leads by a margin of 83%. Comprehensive ablation studies confirm the necessity of each component of WavReward. **All data and code will be publicly after the paper is accepted**.

## 1 Introduction

Spoken dialogue models (Ji et al., 2024a) represent one of the most direct methods of human-computer interaction, evolving from traditional voice assistants such as Alexa, Siri, and Google Assistant to the latest intelligent dialogue systems, such as GPT-4o-audio[1]. Early spoken dialogue models (SpeechTeam, 2024) were typically comprised of automatic speech recognition (ASR) (Cao et al., 2012; Hsu et al., 2021), large language models (LLMs) (Achiam et al., 2023; Touvron et al., 2023; Bai et al., 2023), and text-to-speech (TTS) (Ren et al., 2020; Kong et al., 2023; Shen et al., 2023) components, which facilitated dialogue through a text-based cascading process that bridged speech input and output. To reduce latency and mitigate the cumulative errors of cascading systems, understand and generate non-textual paralinguistic information (e.g., emotion and sound) for real-time interaction, end-to-end spoken dialogue models (Xie & Wu, 2024a; Fang et al., 2024; Xie & Wu, 2024b; Wang et al., 2024b; Chen et al., 2024a; 2025b) such as GPT-4o-audio and Moshi (Défossez et al., 2024) have attracted considerable attention in both academic research and industry. By leveraging vast amounts of speech data (He et al., 2025; Kahn et al., 2020) for multi-stage training, these end-to-end spoken dialogue models (Chen et al., 2025b; Défossez et al., 2024; Xu et al., 2025) not only retain the intelligence quotient of text-based language models but also exhibit **emotional quotient**, they are capable of handling diverse speech-related dialogue scenarios, such as role-playing, emotional dialogue, paralinguistic understanding, and paralinguistic controllable generation.

End-to-end spoken dialogue models (Chen et al., 2025b; Défossez et al., 2024) have demonstrated remarkable conversational abilities, validating the potential of the speech modality in advancing toward Artificial General Intelligence. Thus, assessing the intelligence quotient and emotional

---

[1]https://openai.com/index/chatgpt-can-now-see-hear-and-speak/

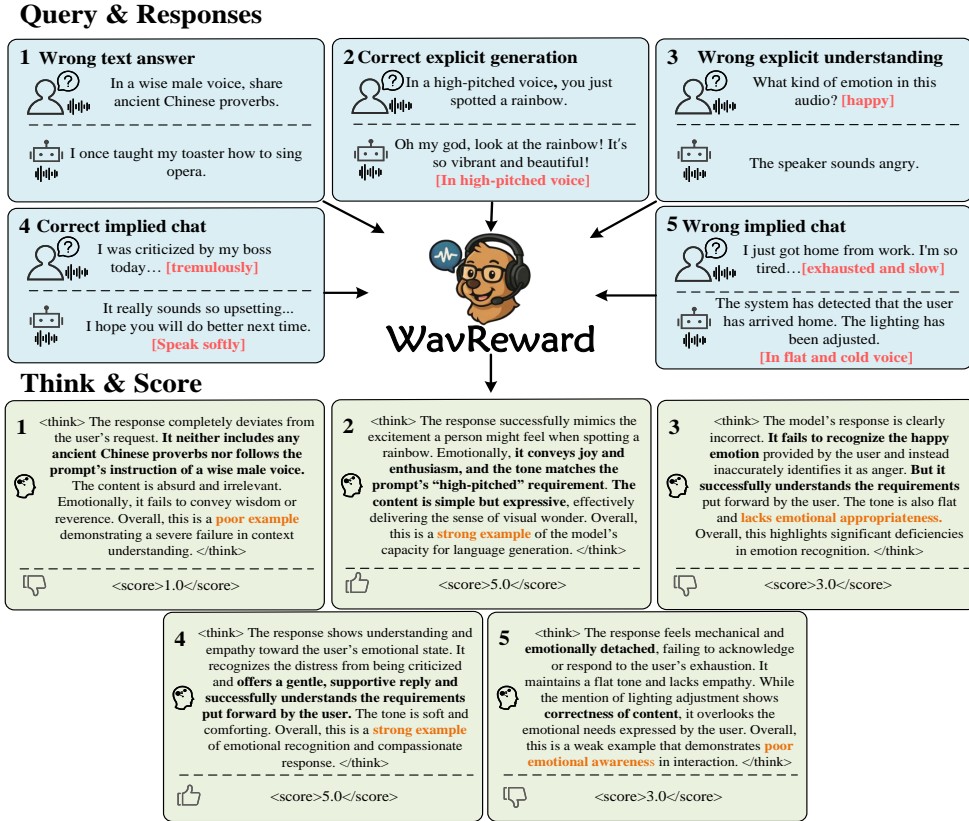

Figure 1: WavReward can be applied to evaluate various dialogue scenarios, including both explicit instruction and implicit dialogues. It directly accepts speech-to-speech dialogue as input, evaluating the conversational coherence at both the textual and acoustic levels, and providing the final score.

quotient of end-to-end spoken dialogue models is a key challenge. This evaluation task involves three main challenges: 1) **the understanding and generation of substantial non-textual acoustic information (e.g., emotion, accent, pitch and sound) often present in dialogue scenarios**, which is currently not well-supported by any dedicated evaluation datasets of dialogue benchmarks (Chen et al., 2024b; Ao et al., 2024; Cheng et al., 2025; Yang et al., 2024). 2) **Dialogue is inherently multi-dimensional and multi-label.** For example, responses from spoken dialogue models may vary in speech rate either faster or slower, without a singular correct answer during casual conversations. 3) **Non-textual information in dialogue is often implicit**. For instance, when user return home late exhausted after work, an intelligent spoken dialogue model should be able to recognize the fatigue from user's voice and respond with a gentle, empathetic tone. Current benchmarks for evaluating spoken dialogue models, such as VoiceBench (Chen et al., 2024b), AirBench (Yang et al., 2024), VoxDialogue (Cheng et al., 2025) and SD-Eval (Ao et al., 2024) primarily focus on the accuracy of textual information in dialogue, similar to using models like ChatGPT to assess the coherence of conversational text. Evaluation of non-textual information is limited to fixed tasks, such as emotion classification, gender recognition, and audio event detection, which assess the model's **understanding of the acoustic information** in the dialogue.

To address the gap in evaluating end-to-end spoken dialogue models, we propose the WavReward model and the ChatReward-30K dataset. WavReward is a novel framework where audio language models (Chu et al., 2024; Xu et al., 2025) (speech-to-text) can serve as evaluators for end-to-end spoken dialogue models (Xie & Wu, 2024a; Défossez et al., 2024). As shown in Figure 1, WavReward can directly assess the capabilities of spoken dialogue models in both textual and non-textual acoustic dimensions. We demonstrate that fine-tuning audio language models with multiple examples via reinforcement learning (Rafailov et al., 2023; Schulman et al., 2017; Shao et al., 2024; Li et al., 2025) enables WavReward to provide reasonable scores across various scenarios. Furthermore,

incorporating chain-of-thought reasoning (Wei et al., 2022; Xie et al., 2025; Ma et al., 2025) into the evaluation process of audio language models significantly aids WavReward in generating more accurate scores. To augment the discriminative capability of WavReward across diverse dialogue contexts, WavReward includes the nonlinear reward mechanism and the positive-negative multi-sample sampling mechanism in the post-training reinforcement learning phase. Additionally, we construct the ChatReward-30K dataset to train WavReward and evaluate the performance of various evaluators (Tang et al., 2023; Chu et al., 2024). ChatReward-30K not only contains standard text-centric dialogue examples but also incorporates diverse acoustic information[2] from end-to-end dialogues. Each speech-to-speech dialogue sample in ChatReward-30K includes multiple responses to the same query. To our knowledge, this is the first dataset capable of comprehensively evaluating both the acoustic capabilities and the implicit conversational abilities of end-to-end spoken dialogue systems. Compared to the original audio language models and the supervised finetuned evaluators, WavReward significantly outperforms these baselines in both in-domain and out-domain scenarios. Furthermore, in human subjective A/B tests, WavReward outperforms direct inference with Qwen2.5-Omni (Xu et al., 2025) by the margin of 83%. In summary, our contributions are as follows:

- WavReward is the first reward model specifically designed for end-to-end spoken dialogue models. It accepts **speech-to-speech dialogues** as input and provides corresponding scores for a wide range of dialogue scenarios. WavReward demonstrates that **audio language models can serve as effective evaluators for spoken dialogue models.**

- WavReward further enhances the evaluative capability through the reasoning-based assessment process, nonlinear reward feedback, and the positive-negative diverse sample sampling mechanism during the reinforcement learning post training.

- We introduce ChatReward-30K, the first dataset designed for training and evaluating audio reward models. Compared to previous datasets, ChatReward-30K enables **comprehensive evaluation of both the acoustic information and implicit dialogue capabilities.**

## 2 RELATED WORK

### 2.1 SPOKEN DIALOGUE MODELS

Spoken dialogue models refer to large language models (Bai et al., 2023; Touvron et al., 2023) capable of engaging in conversations through both speech input and speech output. Traditional spoken dialogue models, such as AudioGPT (Huang et al., 2024) and FunAudioLLM (SpeechTeam, 2024), typically employ a three-stage cascading approach to facilitate dialogue. In this process, speech input is first transcribed into text using an automatic speech recognition model (Cao et al., 2012). The transcribed text is then processed by a text-based LLM such as ChatGPT, to generate a textual response, which is subsequently converted back into speech using a text to speech model (Du et al., 2024a;b). However, these cascaded models often suffer from issues such as high latency, cumulative errors, and an inability to process non-textual acoustic information, which limits their effectiveness. Consequently, end-to-end spoken dialogue models (Défossez et al., 2024; Zhang et al., 2023; Fang et al., 2025) have garnered significant attention in recent months. These models eliminate the need for transcription into text and directly process speech using either semantic (Cao et al., 2012; Hsu et al., 2021; Du et al., 2024a) or acoustic representations (Défossez et al., 2022; Ji et al., 2024b) for understanding and generation. For instance, LLaMA-Omni (Fang et al., 2024) utilizes a Whisper encoder combined with an adapter to process speech, and generates corresponding Hubert tokens based on the LLM, which are then upsampled to produce speech. IntrinsicVoice (Zhang et al., 2024b) introduces GroupFormer to optimize the structure of Hubert token generation, while Mini-Omni1/2 (Xie & Wu, 2024a;b) employs a delay-pattern approach (Copet et al., 2023) to directly generate the corresponding SNAC (Siuzdak et al., 2024) acoustic tokens. Other similar end-to-end spoken dialogue models include SLAM-Omni (Chen et al., 2024a), Freeze-Omni (Wang et al., 2024b), VITA1.5 (Fu et al., 2025), OpenOmni (Luo et al., 2025). Concurrently, numerous end-to-end spoken dialogue models such as GLM-4-Voice (Zeng et al., 2024), Moshi (Défossez et al., 2024), Qwen2.5-Omni (Xu et al., 2025), MinMo (Chen et al., 2025b), Kimi-Audio (Ding et al., 2025), Step-Audio2 (Wu et al., 2025) have demonstrated significant intelligence quotient and emotional quotient emerging from large-scale speech training datasets. Although these spoken dialogue models

---

[2]gender, age, language, accent, pitch, speed, volume, emotion and audio

exhibit strong conversational performance, **there remains a substantial gap in the assessment of both intelligence quotient and emotional quotient. In this paper, we present the first reward model WavReward specifically designed for the evaluation of spoken dialogue models**.

## 2.2 BENCHMARK FOR SPOKEN DIALOGUE MODELS

Early benchmarks related to spoken language, such as AudioBench (Wang et al., 2024a), SU-PERB (Yang et al., 2021), and MMAU (Sakshi et al., 2024), primarily focus on evaluating fixed tasks such as emotion recognition, and are not well-suited for assessing a model's conversational abilities. With the rapid development of end-to-end spoken dialogue models (Défossez et al., 2024; Chen et al., 2025b), numerous new benchmarks have emerged to evaluate these spoken dialogue models. AirBench (Yang et al., 2024) leverages ChatGPT to evaluate the differences between generated text of speech-to-text dialogue models (Chu et al., 2024) and ground truth text at the text level. Spoken-WOZ (Si et al., 2023) transcribes the audio of the conversation into text via ASR models, and then uses metrics like BLEU to assess the performance of text-based language models. VoiceBench (Chen et al., 2024b) transcribes the dialogue audio from speech-to-speech dialogue models (Défossez et al., 2024; Xie & Wu, 2024a) into text and utilizes ChatGPT to evaluate the models' general knowledge and instruction-following ability. VoxDialogue (Cheng et al., 2025) and SD-Eval (Ao et al., 2024) further focus on the ability of speech-to-text dialogue models (Chu et al., 2023; 2024) to understand paralinguistic information, using BLEU and other text-based metrics in conjunction with ChatGPT to assess whether speech-to-text dialogue models (Tang et al., 2023) can generate different textual responses based on varying acoustic information from different users.

However, **the aforementioned benchmarks still rely on transcribing audio into text for evaluation and cannot directly assess the acoustic coherence in speech-to-speech dialogues**. For example, when a user returns home tired after a long day, and the spoken dialogue model responds with a cheerful tone mocking the user, "mocking with a cheerful tone" cannot be directly evaluated by text-based models such as ChatGPT. **WavReward is the first evaluation model that accepts speech input and can directly assess the acoustic dialogue between the user and the spoken dialogue model. It can handle a diverse range of acoustic information, multi-label scenarios, and implicit dialogues scenarios, directly evaluating the realism of the acoustic interactions**. In addition, ChatReward-30K is the first comprehensive dataset **supporting the evaluation of paralinguistic understanding and generation, as well as implicit dialogue scenarios**.

## 2.3 AUDIO LANGUAGE MODELS AS EVALUATORS

It is worth noting that some recent work has attempted to use audio language models as evaluators. For instance, QualiSpeech (Wang et al., 2025) proposed a dataset for speech quality scoring (including the scoring process) and fine-tuned SALMONN using this data. However, WavReward's focus is on the dialogue scenario (semantic and acoustic) for spoken dialogue models, rather than mere speech quality assessment. Furthermore, WavReward is not a simple Supervised Fine-Tuning (SFT) approach; its overall framework is based on a more efficient Reinforcement Learning (RL) method, and it introduces the deep reasoning process (constrained by a separate think component), the nonlinear reward mechanism, and the multi-sample feedback mechanism. Additionally, recent work (Chen et al., 2025a) has also attempted to use audio language models as MOS evaluators, proposing the ALLD method (which aligns the generated sequence of the audio LLM to a text LLM response and uses DPO fine-tuning). WavReward's training process differs significantly from the proposed ALLD method. Moreover, WavReward emphasizes performance in dialogue scenarios (both semantic and acoustic tasks), focuses on out-of-domain generalization, and is committed to open-sourcing the relevant code and data to advance the community.

## 3 METHOD

### 3.1 WAVREWARD

As shown in Figure 2, WavReward is an audio language model (Xu et al., 2025) that undergoes post-training through reinforcement learning (Rafailov et al., 2023; Schulman et al., 2017; Shao et al., 2024; Zhang et al., 2024a). In contrast to text-based large language models (LLMs) such as ChatGPT,

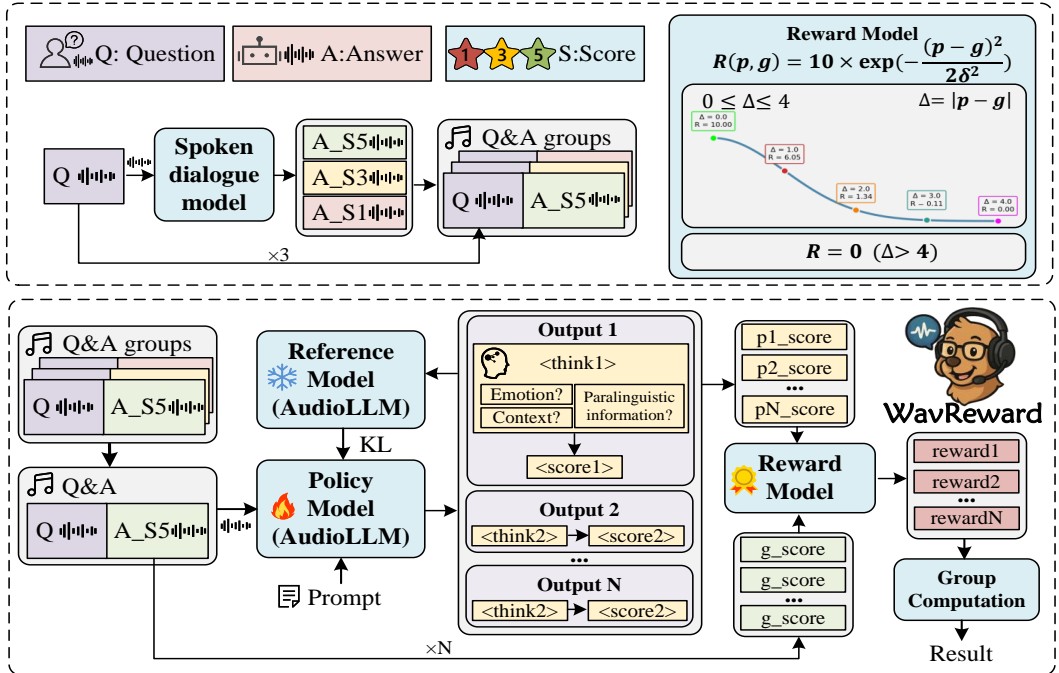

Figure 2: The overall structure of WavReward. WavReward directly accepts speech-to-speech dialogue audio for evaluation. The architecture is based on the audio language model and is trained using reinforcement learning on group samples. Additionally, WavReward incorporates the Chain-of-Thought reasoning process (the center of the diagram), along with positive and negative multi-sample sampling in the top-left corner, and the nonlinear reward mechanism in the top-right corner.

audio language models (Chu et al., 2024; Xu et al., 2025) can directly accept speech-to-speech dialogue as input, enabling a comprehensive evaluation of the coherence of both textual content and acoustic information in explicit and implicit dialogue scenarios. Similar to the conclusions drawn from reinforcement learning in text-based LLMs (Shao et al., 2024), we find that fine-tuning with a small number of precise dialogue scoring samples via reinforcement learning significantly outperforms direct supervised fine-tuning. The relevant ablation results are presented in Table 1.

In the reward models of text-based LLMs, the primary task is to assess whether the content of question-answer pairs is reasonable, typically by sampling and providing feedback based on a single QA sample. However, in the speech dialogue, both the input and output contain abundant content and complex acoustic information. Single-sample QA feedback is insufficient for the reward model to effectively compare differences at various levels(content and acoustic). Therefore, **we design a positive-negative multi-sample feedback mechanism in WavReward**, as shown in the top-left corner of Figure 2. For each dialogue scenario, we construct multiple answer-score pairs $\{a_j, s_j\}$ at different levels for the given question $q$. The first level $s_1$ represents the content of answer that is deemed unreasonable and receives the lowest score. The second level $s_2$ evaluates the acoustic mismatch (e.g., when the user requests the spoken dialogue model to introduce U.S. history in a happy tone, but the model responds in an angry tone). Only when both the content and the acoustic information are correct will the dialogue receive the highest score $s_3$. Therefore, the input $x$ and target $y$ for WavReward during the training process are as follows:

$$x = concat\,(q, a_j)\,, \quad y = s_j, \quad 1 \le s_j \le 5, \quad j \in \{1, 2, 3\} \tag{1}$$

Upon receiving the speech input $x$, WavReward initializes two policy models $W_\theta$ and $W_\theta^{'}$ with identical structures. Both $W_\theta$ and $W_\theta^{'}$ are speech-to-text audio language models (Xu et al., 2025), where $W_\theta^{'}$ serves as the old policy model with frozen weights and the weights of the current training policy model $W_\theta$ remain updatable. Following the approach of DeepSeekMath (GRPO) (Shao et al., 2024), we employ the Kullback-Leibler divergence loss to directly constrain the relationship between

the reference policy model $W_\theta^{ref}$ and the current training policy model $W_\theta$ during the early stages of training. Notably, the KL divergence loss $\mathcal{L}_{KL}(W_\theta, W_\theta^{ref})$ is not incorporated into the reward process of WavReward. The formulation is expressed as follows:

$$\mathcal{L}_{KL}(W_\theta, W_\theta^{'}) = \frac{W_\theta^{ref}(o_{i,t}|x, t_{prompt}, o_{i,<t})}{W_\theta(o_{i,t}|x, t_{prompt}, o_{i,<t})} - log\frac{W_\theta^{ref}(o_{i,t}|x, t_{prompt}, o_{i,<t})}{W_\theta(o_{i,t}|x, t_{prompt}, o_{i,<t})} - 1 \quad (2)$$

where $t_{prompt}$ represents the text prompt for the policy model with specific examples provided in Appendix E, $t$ denotes the number of tokens, $o_i$ refers to the set of $N$ candidate outputs $\{o_1, o_2, \ldots, o_N\}$ sampled by WavReward from the old policy model $W_\theta^{'}$ for each input $x$. It is important to note that each $o_i$ in WavReward is not solely a score for evaluation. **We further incorporate a deep reasoning process by calculating the think format reward $R_f$ (returning 5 or 0 based on compliance), which implicitly enables WavReward to analyze whether the responses $a_i$ of spoken dialogue models address the input question $q$ effectively from both content and acoustic perspectives,** and subsequently assign a final score $p$. WavReward computes the candidate rewards $\{r_1, r_2, \ldots, r_N\}$ for $N$ candidate outputs by comparing the $N$ candidate scores $\{p_1, p_2, \ldots, p_N\}$ with the ground truth score $g$ using the accuracy reward $R_a$. Considering the discrepancy between the acoustic and content information in speech dialogues (the challenge of accurately perceiving acoustic information and providing responses with appropriate acoustic features as compared to content accuracy), **we design a nonlinear accuracy reward $R_a$, as illustrated in the upper-right corner of Figure 2. When the difference between candidate score $p$ and ground score $g$ increases, the reward $R_a$ decreases exponentially, encouraging WavReward to provide higher accuracy rewards to spoken dialogue models that exhibit both cognitive intelligence quotient and emotional quotient.** The explicit formulation of $R_a$ is as follows:

$$R_a(p, q) = \begin{cases} 10 \cdot \exp\left(-\frac{(p-g)^2}{2\sigma^2}\right) & 0 \le |p - g| \le 4 \\ 0 & |p - g| > 4 \end{cases} \quad (3)$$

After obtaining $N$ candidate accuracy rewards $\{r_1, r_2, \ldots, r_N\}$ through $R_a$, WavReward normalizes these accuracy rewards $r_i$ using the mean and standard deviation to derive the corresponding $A_i$:

$$A_i = \frac{r_i - \frac{1}{N}\sum_{i=1}^{N} r_i}{\sqrt{\frac{1}{N}\sum_{i=1}^{N}(r_i - \frac{1}{N}\sum_{i=1}^{N} r_i)^2}} \quad (4)$$

where $A_i$ represents the advantage of the candidate output score $p_i$ relative to other sampled output. Following Guo et al. (2025); Liu et al. (2024a;b); Shao et al. (2024), WavReward encourages the model to generate responses with higher advantages within the group $N$ by updating the policy model $W_\theta$ using the following objective $\mathcal{J}_{WavReward}(\theta)$, where $\epsilon$ and $\beta$ are hyper-parameters:

$$\mathcal{J}_{WavReward}(\theta) = \mathbb{E}[x \sim P(X), \{o_i\}_{i=1}^N \sim W_\theta^{'}(O|x)]$$

$$\frac{1}{N}\sum_{i=1}^{N}\frac{1}{|o_i|}\sum_{t=1}^{|o_i|}\left\{\min\left[\frac{W_\theta(o_{i,t}|x, o_{i,<t})}{W_\theta^{'}(o_{i,t}|x, o_{i,<t})}A_{i,t}, \text{clip}\left(\frac{W_\theta(o_{i,t}|x, o_{i,<t})}{W_\theta^{'}(o_{i,t}|x, o_{i,<t})}, 1-\epsilon, 1+\epsilon\right)A_{i,t}\right]\right.$$

$$\left. - \beta \mathrm{D}_{KL}[W_\theta||W_\theta^{ref}]\right\}$$

$$(5)$$

## 3.2 CHATREWARD-30K

### 3.2.1 THE OVERALL OF CHATREWARD-30K

Given the absence of end-to-end dialogue datasets incorporating scores, we have developed and made available a dataset called ChatReward-30K, which contains spoken dialogue data across various scenarios along with corresponding scores. As shown in Table 3 in Appendix B, ChatReward-30K demonstrates comprehensive coverage compared to existing evaluation datasets (Cheng et al., 2025; Ao et al., 2024) for spoken dialogue models in the following key areas. **1) Evaluation from both**

**content and acoustic dimensions.** Unlike previous datasets (Chen et al., 2024b), ChatReward-30K evaluates dialogue performance from both content and acoustic perspectives, encompassing a wide range of paralinguistic features, including gender, age, language, accent, pitch, speed, volume, energy, emotion and audio. **2) Inclusion of both understanding and generation.** Previous datasets like Voxdialogue and SD-Eval primarily focus on the understanding component (speech-to-text) of spoken dialogue systems. In contrast, ChatReward-30K also evaluates the generation component, providing scenarios that assess how dialogue models generate speech in specific tones, such as speaking in the sad manner. **3) End-to-end implicit dialogue inclusion.** To further assess the emotional intelligence of spoken dialogue models, ChatReward-30K includes implicit dialogues across a variety of scenarios. For instance, it includes a scenario where a voice assistant offers gentle, empathetic comfort at a slow speech rate when the user is crying due to criticism from their boss. **4) Inclusion of both positive and negative examples.** To better train the WavReward model, as outlined in Equation 1, ChatReward-30K provides both positive and negative dialogue responses for the same user scenario. **5) Human expert scoring.** Each dialogue scenario in ChatReward-30K is accompanied by human expert ratings, ensuring that the scores reflect reasonable assessments of dialogue.

### 3.2.2 DATASET STATISTICS

ChatReward-30K consists of the total of 30K samples, each dialogue sample represents the simulated user-chatbot interaction in the form of the speech-to-speech pair. Each dialogue is rated by human experts on a scale from 1 to 5, with the duration of each dialogue audio ranging from 5 to 35 seconds. ChatReward-30K is primarily divided into four components. 15% of the ChatReward-30K focuses on the textual aspects of the conversation. Another 25% of ChatReward-30K addresses the explicit understanding of paralinguistic features such as recognizing when a child is interacting with the spoken dialogue model. The remaining 35% of ChatReward-30K pertains to the model's generation ability of paralinguistic features such as adjusting the volume of the model's voice upon user request. The final 25% represents implicit conversational scenarios such as the spoken dialogue model's ability to automatically detect the user's emotional state and respond appropriately. Detailed examples can be found in Appendix A.

### 3.2.3 DATASET CONSTRUCTION PROCESS

**Stage1: Dialogue Text Generation.** We begin by utilizing the GPT-4 (Achiam et al., 2023) to generate the text portion of the ChatReward-30K dataset through prompt engineering (Reynolds & McDonell, 2021). To ensure the diversity of the dialogue content, we dynamically embed various topics, such as *daily life, health management, education, entertainment, family relations, dietary culture, healthcare, shopping, internet usage, fitness, career development, and social interaction* during the text generation process. To generate explicit instruction-based dialogue data, we instruct the language model to generate dialogues that **contain various metalinguistic information**. For implicit dialogue data, we require the language model to annotate the generated conversation texts **with associated metalinguistic labels**. Prompt programming templates can be found in Appendix D.

**Stage2: Dialogue Speech Generation.** In the generation process, we carefully tailor the most suitable SOTA TTS models for each attribute. We designed customized voice dialogue synthesis pipelines for each attribute to ensure the synthesized dialogue data accurately matches the corresponding attributes: **1) Accent, Pitch, and Emotion:** we utilize GPT-4o-mini-TTS to generate conditionally based speech by adjusting stylistic instructions. This tool focuses on speech techniques such as tongue-twisting, pauses, breathing, and whispering to accurately produce accents and emotions. Based on ten built-in speaker timbres, the model is instructed to synthesize speech using the following command format: Repeat this sentence with the <emotion>/<accent>/<pitch> of <example>. **2) Age:** we randomly selected 1000 speaker (Hechmi et al., 2021) samples from four age groups as reference voices. To minimize textual content discrepancies across different cloned voices, we selected cloned samples with different tones but identical dialogue content for four distinct age groups and used Step-Audio-TTS-3B (Huang et al., 2025) for voice cloning. **3) Speed, Volume, Gender, and Language:** we use CosyVoice2 (Du et al., 2024b) to synthesize speech with specified voice characteristics. The volume and speech rate are adjusted using correlation coefficients to achieve the desired attributes. **4) Audio:** we combine instruct speech clips with audio clips together. Specifically, we selected 39 categories from the AudioCaps (Kim et al., 2019) that include various audio events. After synthesizing all speech segments, we concatenate the simulated user speech segments with the simulated model response speech segments, ensuring a 1-second silence gap between them.

**Step3: Data Filtering and Scoring.** We used the Whisper-Large-V3 (Radford et al., 2023) model to filter out all sentences with the WER greater than 5%. Given the large volume of emotional speech and the ambiguity in category boundaries, we utilized the Emotion2Vec (Ma et al., 2023) model to filter out audio with inaccurate emotional labels and removed synthetic speech with scores below 0.5. To further improve the quality of the ChatReward-30K dataset, we invited five human experts to manually verify and adjust the text, speech, and scoring results of the dataset.

# 4 EXPERIMENTS

## 4.1 EXPERIMENT SETUP

**Datasets.** Since there is currently no dataset available for training and evaluating audio reward model, we use ChatReward-30K-train as the training set for WavReward and evaluate the models using the ChatReward-30K-test (4000 samples) across three aspects: **content, explicit paralinguistic understanding and generation (with 9 distinct paralinguistic features), and implicit dialogue**. Additionally, we record 120 real human-machine dialogues between users and LLaMA-Omni (Fang et al., 2024) (overall biased negative samples) and Kimi-Audio (Ding et al., 2025) (overall biased positive samples), named the RealDialogue **to compare the performance of different evaluators in more realistic out-of-domain settings**. In the RealDialogue dataset, we observed that certain dialogues have extended durations, and there are instances of poor audio quality, such as distorted electronic sounds. **These factors present a more rigorous challenge for evaluating the model's performance in unseen, real-world scenarios.**

**Baselines.** Similar to using ChatGPT for assessing the coherence of text-based dialogues, we employ various audio language models (Chu et al., 2024) (speech-to-text) as baseline evaluators to score speech-to-speech dialogues. The specific audio language models include Qwen-Audio (Chu et al., 2023), SALMONN (Tang et al., 2023), Audio Flamingo2 (Ghosh et al., 2025), Qwen2-Audio (Chu et al., 2024), Qwen2.5-Omni (Xu et al., 2025) and GPT-4o-audio. Furthermore, we enhance two new versions by fine-tuning Qwen2.5-Omni using both full-parameter and LoRA (Hu et al., 2022) fine-tuning methods on the WavReward-30K-train with ms-swift (Zhao et al., 2025). Training details and Metrics are in Appendix F

## 4.2 MAIN RESULTS

Table 1: The **accuracy** of scoring by WavReward and various baselines on the ChatReward-test and RealDialogue datasets is evaluated. Specifically, the ChatReward-test dataset is assessed across three main dimensions: content scoring, acoustic instruction dialogue scoring (which includes both understanding and generation), and implicit dialogue scoring. The acoustic information which are categorized as follows: age, accent (acc.), gender (gen.), language (lan.), emotion (emo.), volume (vol.), speech rate (spe.), pitch (pit.), and audio (aud.).

| Model | Content | Acoustic Instruction | | | | | | | | | Implicit | RealDialogue |
|---|---|---|---|---|---|---|---|---|---|---|---|---|
| | | age. | acc. | lan. | gen. | emo. | pit. | spe. | vol. | aud. | | |
| *I. Baseline audio language models direct inference with prompt* | | | | | | | | | | | | |
| Qwen2-Audio | 24.7 | 32.4 | 24.5 | 36.8 | 27.6 | 33.7 | 32.3 | 40.5 | 41.4 | 50.8 | 28.9 | 42.5 |
| Qwen-Audio | 43.4 | 35.5 | 23.0 | 33.6 | 14.9 | 34.2 | 27.7 | 35.4 | 39.0 | 32.2 | 35.2 | 40.8 |
| SALMONN | 13.5 | 33.9 | 34.8 | 28.4 | 33.9 | 36.4 | 25.4 | 30.3 | 28.0 | 51.3 | 20.3 | 19.2 |
| Audio Flamingo2 | 22.7 | 20.8 | 25.4 | 16.8 | 18.8 | 18.6 | 17.8 | 21.5 | 21.9 | 20.6 | 22.6 | 21.6 |
| GPT-4o-audio | 69.4 | 92.0 | 57.7 | 100 | 82.1 | 58.5 | 88.7 | 94.5 | 88.1 | 83.3 | 53.6 | 57.6 |
| Qwen2.5-Omni | 54.6 | 56.3 | 50.0 | 48.4 | 54.1 | 57.8 | 66.1 | 34.1 | 53.6 | 64.9 | 48.5 | 51.7 |
| *II. Baseline audio language models after supervised fine-tuning* | | | | | | | | | | | | |
| Qwen2.5-Omni w/ Full-param tuning | 67.1 | 81.9 | 69.6 | 98.9 | 83.8 | 66.5 | 76.1 | 84.8 | 86.6 | 86.7 | 55.8 | 58.3 |
| Qwen2.5-Omni w/ LoRA | 63.8 | 81.2 | 43.6 | 100 | 82.1 | 49.1 | 74.1 | 83.6 | 85.1 | 85.7 | 54.2 | 56.7 |
| *III. Different ablation versions of WavReward* | | | | | | | | | | | | |
| WavReward w/o cot think | 84.2 | 80.3 | 77.9 | 98.9 | 86.9 | 85.4 | 80.7 | 86.0 | 90.2 | 88.6 | 61.4 | 59.1 |
| WavReward w/o multi samples | 85.3 | 85.7 | 69.6 | 98.9 | 88.6 | 90.0 | 82.1 | 88.6 | 85.4 | 90.2 | 61.9 | 72.5 |
| WavReward w/o nonlinear reward | 88.6 | 92.2 | 80.9 | 100 | 89.8 | 90.5 | 83.8 | 87.3 | 92.7 | 85.7 | 66.6 | 70.8 |
| **WavReward (ours)** | **90.8** | **96.9** | **87.7** | **100** | **95.5** | **97.5** | **89.1** | **91.1** | **97.6** | **97.0** | **74.3** | **80.8** |

We evaluated the generation and ground-truth score accuracy of WavReward and Baseline models on the ChatReward-30K-test as well as the real out-of-domain RealDialogue dataset. The Baseline is divided into two categories: one consists of direct inference from audio language models using text prompt templates consistent with WavReward, and the other is the evaluator fine-tuned using the ChatReward-30K-train. The specific experimental results are presented in Table 1. We can draw

the following conclusions: **1) WavReward significantly outperforms the best audio language models GPT-4o-audio on all metrics**. It achieved improvements of 21.4, 20.7, and 39.0 points in the content scoring, implicit dialogue scoring, and emotion-instructed dialogue scoring, respectively. Furthermore, it outperformed the direct inference model Qwen2.5-Omni by an average factor of two. This indicates that audio language models when optimized using reinforcement learning, can effectively serve as evaluators for spoken dialogue models. Moreover, RL significantly improves performance compared to direct inference. **2) We found that the RL-based WavReward surpassed the LoRA fine-tuned Qwen2.5-Omni**. This may be due to the direct scoring approach of supervised fine-tuning, which is overly simplistic and struggles to capture the complex scoring logic needed for various evaluation scenarios. **3) We observed a substantial performance gap between different audio language models during direct inference**, highlighting the need for future work to develop and open-source more robust foundational audio language models. **4) On the RealDialogue dataset, WavReward achieved a score accuracy of 80%,** indicating that it exhibits strong robustness and can provide reliable evaluations in real-world, complex scenarios.

### 4.3 A/B Test on RealDialogue

Given that evaluating the responses of end-to-end spoken dialogue models in implicit dialogue settings constitutes multi-label scenario and the responses of dialogue models should ideally align with human preferences, we have incorporated a subjective A/B testing approach. Specifically, five human experts were tasked with evaluating data from RealDialogue and determining which of two distinct discriminators provided the most reasonable assessment. We conducted pairwise comparisons between three baseline: Qwen2.5-Omni w/ direct inference, GPT-4o-audio w/ direct inference, and Qwen2.5-Omni w/ LoRA. The objective was to compare the different scoring outcomes of WavReward and the baseline model on the same sample, with the results presented in Table 2. Our findings indicate that WavReward outperformed Qwen2.5-Omni w/ direct inference in the subjective A/B test by a margin of 83%, and also achieved a 77% success rate when compared to GPT-4o-audio. These results suggest that WavReward is more closely aligned with human preferences, demonstrating superior performance across a wide range of real-world dialogue scenarios.

Table 2: Subjective A/B testing of WavReward and different evaluators on the RealDialogue dataset.

| Models | WavReward Win ↑ | WavReward Lose ↓ |
|---|---|---|
| Qwen2.5-Omni w/ direct inference | 83 | 17 |
| Qwen2.5-Omni w/ LoRA | 79 | 21 |
| GPT-4o-audio w/ direct inference | 77 | 23 |

### 4.4 Ablation experiments

**w/o cot think.** We removed the chain-of-thought (CoT) reasoning from WavReward and referred to this version as WavReward w/o CoT think. Specifically, WavReward in this configuration directly generates scores without the additional CoT-based format reward (loss) and the corresponding $t_{prompt}$ was also modified as shown in Appendix E. All other training and model parameters remain unchanged. As shown in Table 1, we found that CoT reasoning improved accuracy by approximately 10% across all evaluation categories. In out-of-domain scenarios, the improvement was as high as 21.7%. This suggests that reasoning capabilities are beneficial for the evaluator model.

**w/o nolinear reward.** We replaced the reward function in Equation 3 with a classic linear 0/1 reward function (Liu et al., 2024a;b). Specifically, the WavReward w/o nolinear reward version receives the reward of 5 when the generated score matches the ground truth score, and no reward is given when there is a mismatch during training. All other training and model parameters are consistent with the previous configuration. By comparing the versions of WavReward and WavReward w/o nonlinear in Table 1, we observe that the non-linear reward function aids WavReward in learning the differences in various levels of information in speech. For instance, when there is a large discrepancy between the ground truth score and the predicted score (e.g., a high emotional intelligence response receives a low score of 1 from WavReward), a substantial penalty is applied which helps the model correct such errors.

**w/o multi samples.** In classical reinforcement learning algorithms (Schulman et al., 2017), single-sample sampling can be used to calculate rewards based on the difference between the ground truth

score and the generated score. In the WavReward w/o multi-samples version, for each question only one randomly selected answer is used for evaluation. We found that performance dropped when multi-sample evaluation was removed. This decline can be attributed to the loss of the ability to simulate a range of reasonable and unreasonable responses to the same question, which assists WavReward in distinguishing between different scoring criteria and their variations.

## 5 CONCLUSION

In this work, we present WavReward, the first evaluation framework capable of supporting speech-to-speech input and providing comprehensive assessments of spoken dialogue models at both the text and acoustic levels. WavReward leverages reinforcement learning to turn audio language models into evaluatorsn and incorporate the chain-of-thought reasoning process, nonlinear rewards, and both positive and negative sample feedback to enhance the validity of the evaluation. In a variety of in-domain and out-of-domain explicit and implicit evaluation scenarios, WavReward outperforms previous state-of-the-art evaluators. In the future, we aim to scale up audio language models (e.g., 7B-70B) to further enhance WavReward's capabilities.

## ETHICAL CONSIDERATIONS

WavReward was developed with rigorous attention to ethical and responsible research practices. All speech data utilized in this work, including the underlying data for the ChatReward-30K and RealDialogue benchmarks, are either synthetic or no personally identifiable information (PII) is included. To further protect speaker privacy, the data were processed to ensure anonymity and are used exclusively for academic research purposes.

We acknowledge that spoken dialogue corpora, even synthetic ones, may inadvertently reflect existing social, cultural, or gender biases, which could be inherited by models trained or evaluated using them. While WavReward is intended to advance research in robust spoken dialogue evaluation, it should not be regarded as bias-free. We strongly encourage future researchers to critically examine and mitigate any potential harms or biases that may arise from evaluation on this benchmark or the use of the model.

Finally, as large language and speech models become increasingly powerful, they also present risks of misuse, such as generating misleading content or enabling invasive surveillance. Therefore, WavReward is released solely for academic and responsible industrial research, with the overarching goal of fostering dialogue systems that are beneficial, transparent, and aligned with human values.

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

## A  Examples and details in ChatReward-30K

Various examples from the ChatReward-30K dataset are illustrated in Figure 3.

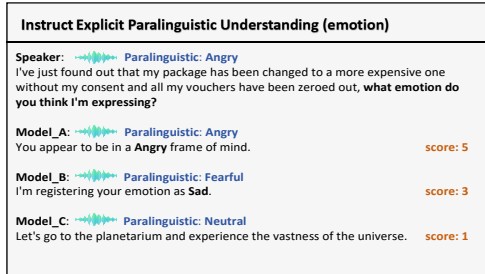

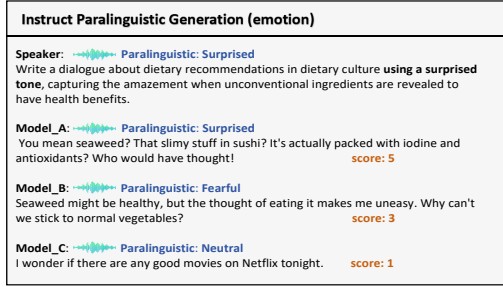

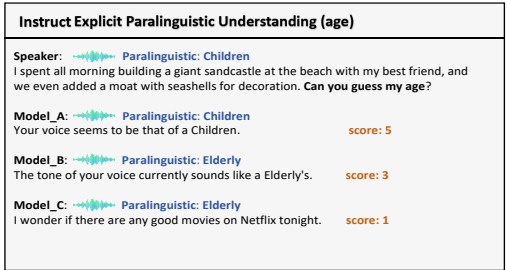

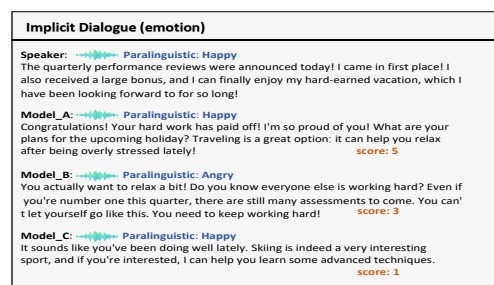

Figure 3: The ChatReward-30K dataset encompasses a wide range of both explicit and implicit dialogue scenarios, with responses evaluated by human experts based on model performance.

Following Equation 1, each dialogue sample contains both positive and negative responses for the same user input. In terms of content, the dialogues in ChatReward are more aligned with natural and daily conversations rather than explicit QA pairs. As shown in Figure 7, the word cloud visualization of ChatReward-30K demonstrates a prevalence of natural spoken words, such as "can't" which

is representative of daily spoken interactions. Concerning the acoustic attributes of the dialogues, most attribute categories in ChatReward-30K exhibit a relatively balanced distribution, as shown in Figure 6. Given the subtle emotional cues that humans can perceive in dialogue models, ChatReward-30K assigns particular emphasis to emotional attributes. Detailed information on each category is provided in Appendix C. The ChatReward is ultimately split into ChatReward-30K-train and ChatReward-30K-test sets with ratios of 85% and 15% respectively.

## B  COMPARISON OF DIFFERENT EVALUATION DATASET/BENCHMARK FOR SPOKEN DIALOGUE MODELS

The comparison is shown is Table 3.

Table 3: Comparison of different evaluation dataset/benchmark for spoken dialogue models. **Dia.** refers to spoken **dialogue** and pure question-answering evaluation is not categorized as the dialogue (chat) task. **S2S.** denotes evaluation of **speech-to-speech** models. **Imp.** indicates **implicit** dialogues. **Neg. and Sco.** represent whether all positive and **negative** samples in the evaluation data are **scored**. Acoustic Information covers aspects like age, accent (acc.), gender (gen.), language (lan.), emotion (emo.), volume (vol.), speech rate (spe.), pitch (pit.) and audio (aud.).

| Dataset/Benchmark | Dia. | S2S. | Neg. | Imp. | Sco. | age. | acc. | lan. | gen. | emo. | pit. | spe. | vol. | aud. |
|---|---|---|---|---|---|---|---|---|---|---|---|---|---|---|
| | | | | | | **Acoustic Paralinguistic Information** | | | | | | | | |
| SUPERB | ✗ | ✗ | ✗ | ✗ | ✗ | ✗ | ✗ | ✗ | ✗ | ✓ | ✗ | ✗ | ✗ | ✗ |
| MMAU | ✗ | ✗ | ✗ | ✗ | ✗ | ✗ | ✗ | ✗ | ✗ | ✓ | ✗ | ✗ | ✗ | ✓ |
| AudioBench | ✗ | ✗ | ✗ | ✗ | ✗ | ✗ | ✓ | ✗ | ✓ | ✓ | ✗ | ✗ | ✗ | ✓ |
| AirBench | ✓ | ✗ | ✗ | ✗ | ✗ | ✗ | ✗ | ✗ | ✓ | ✓ | ✗ | ✗ | ✗ | ✓ |
| SD-Eval | ✓ | ✗ | ✗ | ✗ | ✗ | ✓ | ✓ | ✗ | ✓ | ✓ | ✗ | ✗ | ✗ | ✓ |
| VoiceBench | ✓ | ✓ | ✗ | ✗ | ✗ | ✗ | ✗ | ✗ | ✗ | ✗ | ✗ | ✗ | ✗ | ✗ |
| VoxDialogue | ✓ | ✗ | ✗ | ✓ | ✗ | ✓ | ✓ | ✓ | ✓ | ✓ | ✓ | ✓ | ✓ | ✓ |
| ChatReward-30K | ✓ | ✓ | ✓ | ✓ | ✓ | ✓ | ✓ | ✓ | ✓ | ✓ | ✓ | ✓ | ✓ | ✓ |

## C  ACOUSTIC INFORMATION IN CHATREWARD-30K

The specific categories, sample quantities, and durations of all acoustic information in ChatReward-30K are detailed in Table 4.

Table 4: Detailed statistics of the corresponding subsets of each attribute in ChatReward-30K.

| Attributes | Categories | Samples | Duration |
|---|---|---|---|
| Gender | male, female | 2177 | 9.56Hours |
| Age | children, elderly, middle-aged, adolescent | 2070 | 8.36Hours |
| Language | chinese, english | 3583 | 16.23Hours |
| Accent | indian, canadian, british, singaporean, american, australian | 1618 | 5.70Hours |
| Emotion | neutral, happy, sad, angry, surprised, disgusted, fearful | 9470 | 52.04Hours |
| Pitch | low, high, normal | 853 | 3.65Hours |
| Speed | slow, normal, fast | 2303 | 10.95Hours |
| Volume | low, normal, high | 2054 | 7.53Hours |
| Audio | laughing, crying, bee, bird, car, cat, chirping, clapping, coughing, dog, screaming duck, horse, ice, knocking, ocean, pig, police, sneezing, thunder, waterfall burbling | 4081 | 15.38Hours |
| **Overall** | | **28209** | **129.40Hours** |

## D  PROMPT PROGRAMMING TEMPLATE FOR CHATREWARD-30K

Regarding the generation of high-quality implicit dialogue data. To ensure textual diversity, we designed various distinct dialogue scenarios (as detailed in Line 357 and Line 874 of the revised manuscript). To ensure textual accuracy, we incorporated relevant constraints and example parameters into the prompts (as detailed in Line 876 and Line 880). Furthermore, we observed that implicit dialogue scenarios are often more distinctly correlated with emotion, speaking rate, and volume.

Thus, we have augmented the data in these dimensions for the implicit dialogue contexts.When constructing the dialogue dataset with distinctly different scores (3 points vs. 5 points), we specified significantly disparate emotions (e.g., Happy vs. Angry), rather than subtle differences (e.g., Happy vs. Gentle), to ensure a clear distinction between the reward signals. The sample prompt template for ChatReward-30K is illustrated in the Figure 4 below:

---

**Prompt Template for Implicit Dialogue Scenarios in Emotional Types**

Requirements:
You are a professional designer of single-round conversation instructions tasked with creating single-round emotion training instructions.
1. The user's input reflects the scenario '{scenario}' and implies the emotion '{emotion}', but must not directly mention emotion words (e.g., "happy", "sad", "angry", etc.).
2. The model's response should naturally adapt to the implied emotion of the scenario, with a tone close to everyday communication, avoiding imperative or unnatural expressions.
3. Provide an appropriate emotion label ("{emotion}") and an incorrect emotion label (clearly not matching the scenario).
4. Both the right emotional label and the wrong emotional label must belong to {EMOTION_CATEGORIES}
5. User input must be vivid, relevant, and based on the provided scenario.

Example for '{emotion}' emotion:
- User: "User input example for {emotion}",
- Model: "Model response example for {emotion}",
- Appropriate label: "{emotion}",
- Incorrect label: "{random.choice([e for e in EMOTION_CATEGORIES if e != emotion])}"

Please generate a new single-turn dialogue based on the scenario '{scenario}', implying the "{emotion}" emotion, strictly following the format below:
- User: "User input",
- Model: "Model response",
- Appropriate label: "{emotion}",
- Incorrect label: "Incorrect label"

---

Figure 4: The emotion prompt template for ChatReward-30K.

# E    PROMPT PROGRAMMING TEMPLATE (WITHOUT COT) FOR WAVREWARD

The ablation text prompt template for WavReward w/o COT is shown in Figure 5.

---

**Without Chain of Thought Prompt Template for WavReward**

```
Prompt_template = (
    "## Dialogue Response Evaluation\n\n"
    "**IMPORTANT:** Evaluation must include`<score>` rating.\n\n"
    "Listen to the dialogue recording (two sentences, 1-second pause in between). Evaluate the quality of the **second
sentence** as a response to the first, focusing on **text relevance** and the **appropriateness** of **Linguistic information
(a range of paralinguistic information such as emotion/age/pitch/speed/volume)**.\n"
    "**Note:** Focus on evaluating the appropriateness of the second sentence relative to the first, even if the first sentence
itself contains contradictory information.\n\n"

    "## Scoring Criteria\n\n"
    "**1 points**: Text content is irrelevant or incorrect or illogical.(low intelligence)\n"
    "**3 points**: Text is relevant, but paralinguistic information is **inappropriate** for the context.(low emotional
quotient)\n"
    "**5 points**: Text is relevant, and paralinguistic information is **appropriate** for the context, resulting in effective
communication.(High intelligence and emotional intelligence.)\n\n"

    "## Evaluation Requirements\n\n"
    "Response **MUST** follow this format:\n\n"
    "<score>X</score> (**X is 1, 3, or 5**)\n\n")

obj["prompt"] = [{"role": "user", "content": [
    {"type": "audio", "audio": obj["merge_wav"]},
    {"type": "text", "text": Prompt_template}]
}]
```

---

Figure 5: The ablation prompt template for WavReward.

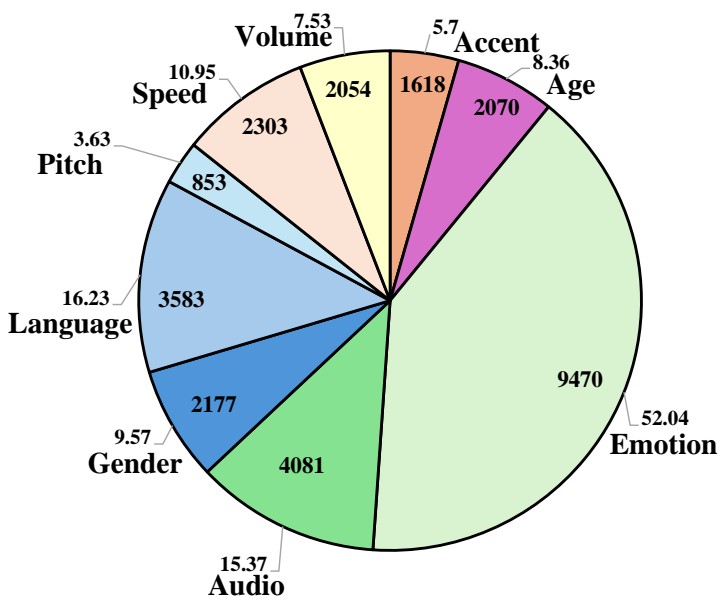

Figure 6: Statistics of different acoustic attribute in ChatReward-30K.

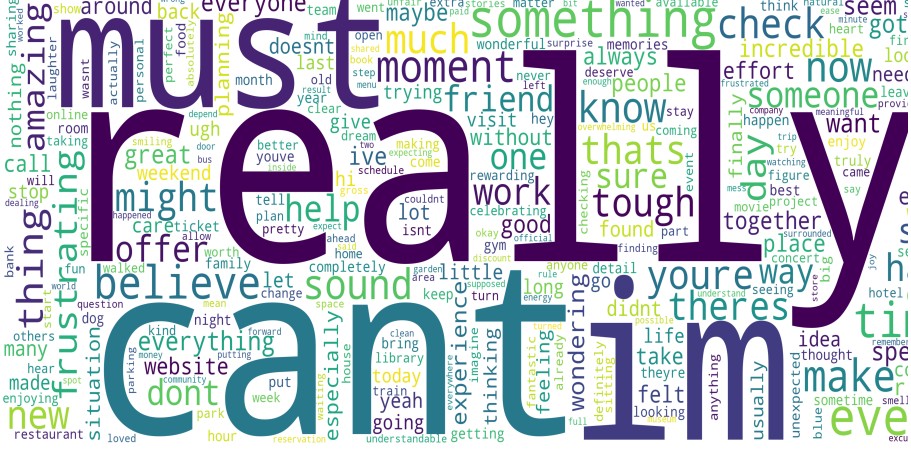

Figure 7: Word Cloud of ChatReward-30K.

## F    TRAINING DETAILS AND METRIC

**Training details.** WavReward is trained using 8 H20 96GB GPUs, each running a batch size of 1, with gradient accumulation performed every 2 steps. The model is trained for 3500 steps with a learning rate of $1 \times 10^{-6}$ and a temperature of 1.0. The maximum number of cot tokens is set to 5120, and the weight coefficient for the KL loss is set to 0.01. The model architecture of WavReward is based on the sota open-source audio language model Qwen2.5-Omni-7B (Thinker) (Xu et al., 2025) with the identical parameters. All parameters of WavReward are updated during the training process. **Metric.** For evaluation on the ChatReward-30K-test and RealDialogue, we use accuracy to measure the difference between the predicted scores and the ground truth scores. On the RealDialogue-testset, we conduct subjective A/B testing via crowdsourcing, where 5 human experts are required to select the optimal score between the different scores given by WavReward and various baseline evaluators in the same real dialogue.

## G    THE USE OF LARGE LANGUAGE MODELS

In this study, we employed Gemini-2.5-Pro (Comanici et al., 2025) to facilitate the comprehensive detection of linguistic inaccuracies within the manuscript, encompassing spelling errors, punctuation misuses, and grammatical irregularities.

## H    COMPUTATIONAL COST

Regarding the latency associated with the Chain-of-Thought mechanism, we conducted further evaluation using the RealDialogue test set on a single H20 96G GPU. We measured the Real-Time Factor (RTF) for both the baseline (WavReward w/o cot think) and the full model (WavReward). RTF is defined as the time taken to generate the score divided by the duration of the input dialogue data. The results shown from Table 5 confirm that while CoT provides a substantial performance gain (as demonstrated in Table 1), it does introduce a certain degree of increased inference latency.

Table 5: RTF of inference.

| Models | RTF $\downarrow$ |
|---|---|
| WavReward w/o cot think | 0.172 |
| WavReward | 0.318 |

## I    CHAIN-OF-THOUGHT DETAILS

we leveraged the advanced Gemini-2.5-Pro model (Comanici et al., 2025) to evaluate the alignment between the input speech dialogue data and the corresponding CoT text, providing a binary classification of Good or Poor quality. The strict requirements imposed by the prompt included the CoT's semantic completeness, its correct analysis from both semantic and acoustic dimensions, and whether it faithfully reflects the genuine reasoning process of the dialogue. We tested the inference CoT text generated by WavReward on the ChatReward-30K-test dataset, with the results shown the accuracy is 96.6%.

