# OpenReview forum: "WavReward: Spoken Dialogue Models With Generalist Reward Evaluators"
_ICLR.cc/2026/Conference — Submitted to ICLR 2026_

### Official Review · Reviewer_8qWV · 2025-10-23

**Soundness:** 2
**Presentation:** 2
**Contribution:** 3
**Rating:** 2
**Confidence:** 4

**Summary:**

This paper proposed a generalist reward model for single-turn spoken dialogues. The authors proposed training the reward model with GRPO based on the IQ and EQ of the spoken response. In addition, they added CoT in the reward model and also an exponential curve design for calculating the reward for GRPO training. They also curated a dataset, ChatReward-30K, for training and evaluating reward models. For out-of-distribution evaluation, they curated the RealDialogue dataset, which records real human interaction with speech models. Throughout their ablation studies, they demonstrated the effectiveness of their training method over baselines with a large margin on their evaluation benchmark.

**Strengths:**

1. It’s the first paper to propose a "generalist" evaluator for spoken dialogue. I believe this contribution will enlighten further research in the field of spoken dialogue systems.
2. Their ablation shows the effectiveness of their proposed methods: CoT/multisamples/nonlinear reward.

**Weaknesses:**

1. The writing of this paper is hard for readers to follow. There are two models that are called reward models in this paper: the proposed WavReward and the reward model used to train WavReward. I strongly recommend that the author change the naming.
2. It’s a concern for me that the authors only invited 5 human experts to manually verify ChatReward-30K, which is an unreasonable workload for ensuring the quality of the dataset on such a large scale..
3. In natural human conversation, there exist multiple reasonable spoken responses (content and paralinguistic-wise) for a given query. I’m questioning whether it is realistic to train and evaluate such evaluators with a ground truth score from a dataset with only 5 human experts involved?

**Questions:**

1. Is there a typo in equation 2? There is no $W_{\theta}^{'}$ in the right-hand side of the equation.
2. If I’m understanding correctly, the authors are using GRPO for their reward model training. Why does the author not explicitly mention GRPO in their paper?
3. In Appendix F, the author mentions that their WavReward is initialized from a model named “Qwen-2.5-Omni-7B-Think”. However, I cannot find this model released on the websites/huggingface. Could the author provide some details on this?
4. I’m not fully understanding the sentence (Line 255 to 257) “Notably, the KL divergence loss $\mathcal{L}\_{KL}{\left( W_{\theta}, W_{\theta}^{ref} \right) }$ is not incorporated into the reward process of WavReward.” What do the authors mean by “reward process”?
5. How is the baseline model trained via supervised fine-tuning? What is the objective?
6. For the A/B testing experiment on the RealDialogue dataset, the authors mentioned that 5 human experts are involved in selecting optimal scores between different models. Are these 5 human experts the same group of people for the manual verification of ChatReward-30K?

**Details Of Ethics Concerns:**

I think there is an ethic concern for the releasing of ChatReward-30K dataset. However, I do not see an ethical statement in the paper.

---

> ### Author Response · Authors · 2025-11-24
> **Rebuttal to Reviewer 8qWV(1/2)**
>
> We sincerely thank you for acknowledging the contributions of WavReward. We recognize that your primary weakness concern revolves around the involvement of five experts in constructing the ChatReward-30K dataset, and we are eager to provide a detailed explanation of the underlying process.
>
> ### **W1: The name of WavReward**
> Thank you for your suggestion. WavReward is designed as a **model-based reward model (evaluator)**. Since we employ Reinforcement Learning (RL) during its training, **the process incorporates a rule-based reward** as part of the RL objective. These two types of rewards belong to different categories, but we appreciate your attention to detail. We initially chose "WavReward" over "WavEvaluator" for the title because our subsequent work intends to fully utilize the WavReward framework within the RL loop of end-to-end dialogue models. Thank you for your understanding.
>
>
> ### **W2/W3: Involvement of Five Experts in ChatReward-30K**
>
> You might be concerned that the **30,000 scores in ChatReward-30K were assigned one by one by human experts**, leading to an overwhelming workload. In reality, human experts were involved in the ChatReward-30K construction process in a more efficient, quality-control manner:
>
> (1). For Explicit Dialogue Scenarios (e.g., the query: "Please use a grandmother's voice to tell me a Greek mythology story"): We first instruct GPT to generate both a correct answer text and an incorrect answer text. The incorrect text is automatically assigned 1 point. For the correct text, we use the corresponding correct paralinguistic label (e.g., elderly female voice) and then **programmatically select a wrong paralinguistic label (e.g., child's voice)**. Running these through the TTS pipeline yields the 5-point and 3-point samples respectively. Thus, human experts only need to verify the correctness/incorrectness of the GPT-generated text. Given GPT's high fidelity in instruction-following, our expert team can accomplish this verification task efficiently. The process is similar for the explicit comprehension part of ChatReward-30K.
>
> (2). For Implicit Dialogue Scenarios: We instruct GPT to generate the corresponding dialogue text (including both query and answer) based on the correct paralinguistic label (e.g., "Happy"). Similarly, an incorrect answer text is generated and assigned 1 point. **Human experts are required to strictly verify whether the implicit dialogue scenario generated by the model aligns with the intended "happy" response. However, the workload for this high-quality subset was manageable, averaging only 400 samples per expert for implicit dialogues with scores not equal to 1**. We consider this workload to be reasonable.
>
> (4) The data in ChatReward-30K is primarily synthetic (text from GPT, speech from TTS, and scores from rule-based classification). TTS verification is only needed for specific, difficult-to-synthesize accents. For the RealDialogue human-machine dataset, we did require human experts to assign scores one by one and justifications, **but given that RealDialogue contains only 120 samples, this workload was also reasonable.**
>
> (5) The human experts are members of our professional LLM data team in our company.
>
> (6) Both ChatReward-30K and WavReward will be open-sourced.

---

> > ### Author Response · Authors · 2025-11-24
> > **Rebuttal to Reviewer 8qWV(2/2)**
> >
> > ### **Q1.**
> > Thank you for your attention. There is no typo. The KL term in the overall loss primarily regularizes the current policy $W_{\theta}$ against the reference policy $W_{\theta}^{ref}$ (the model from step 0, typically the SFT model). In contrast, the $W_{\theta}'$ in Equation (5) refers to the old policy model (the model from the previous step, used for sampling).
> >
> > ### **Q2.**
> > Thank you for your suggestion. We had cited DeepSeekMath in the original paper's main text. We have now further supplemented for GRPO.
> >
> > ### **Q3.**
> > The GitHub repository for Qwen2.5-Omni is continuously being updated. Qwen2.5-Omni includes the "thinker-talker" components, but we only utilized the Thinker part. The specific version we used, **Qwen2_5OmniThinkerForConditionalGeneration**, can be found in the official **transformers github repository in commit 4b8c6d4 (Add Qwen2.5-Omni #36752) README**. We have clarified this in the revised paper.
> >
> > ### **Q4.**
> > In traditional algorithms[1], the KL loss is directly added as a regularization term to the objective function (the reward process in the WavReward paper). However, this method typically requires normalization, sampling, and logarithmic operations for the policy distribution to compute the KL divergence. Performing the calculation of A (Advantage) and KL concurrently complicates the entire objective function. Furthermore, forcefully inserting the absolute KL penalty term at this stage would compromise the effectiveness of the relative optimization framework of GRPO. Therefore, following the methodology of GRPO, our KL loss is added as part of the **final total loss**.
> >
> > ### **Q5.**
> > The baseline models were fine-tuned using the official ModelScope fine-tuning framework ms-swift. The fine-tuning objective was the LLM Cross-Entropy Loss with respect to the corresponding scores in ChatReward-30K.
> >
> > ### **Q6.**
> > No, they are not the same group of people. The human experts are also members of the professional LLM data team in our company.
> >
> > ### **Q7.**
> > We have included the relevant Ethical Considerations at the end of the main paper.
> >
> >
> > **Thank you again for your suggestions! We are very willing to provide any further clarification you may need and eagerly look forward to your continued support!**
> >
> > [1]Training language models to follow instructions with human feedback

---

> > > ### Author Response · Authors · 2025-11-26
> > >
> > > Dear Reviewer 8qWV:
> > >
> > > We sincerely thank you for acknowledging some contributions of WavReward! As we are in the discussion period, we would be grateful if we could hear your feedback regarding our answers to the reviews. We would be happy to answer and discuss if you have further comments. Thanks for your expertise and precious time!
> > >
> > > Best regards.
> > >
> > > The WavReward Authors.

---

> > > > ### Comment · Reviewer_8qWV · 2025-11-28
> > > >
> > > > I thanks the authors for the comprehensive responses and clarifications, which have addressed many of my initial concerns and led me to raise my score.
> > > >
> > > > However, I still have reservations concerning :W2/W3: Involvement of 5 Experts in ChatReward-30K."  I believe the limited number of 5 human experts is insufficient, especially since the dataset includes conversations involving paralinguistic aspects. Unlike objective domains such as math or logical reasoning, these paralinguistic aspects likely involve greater variation in human judgment. To accurately capture the real distribution of human preferences, a larger number of experts is necessary. Without this, I'm not confident that this dataset fully reflects the real human distribution.

---

> ### Author Response · Authors · 2025-12-02
> **Further Discussion with Reviewer 8qWV**
>
> Thank you very much for your valuable feedback and generous support for our work on WavReward. **We deeply appreciate your commitment to enhancing the score.** Regarding your final point that filtering the ChatReward-30K dataset with only five human experts is insufficient, **we regret to respectfully state that we hold a contrasting view.**
>
> In the domain of paralinguistic attributes (including age, gender, language, accent, pitch, speed, energy, emotion, and audio), several features are easily synthesized and reliably judged by human subjects, often achieving high synthesis accuracy in TTS models. For instance, attributes like language (e.g., Chinese vs. English) and speaking rate (e.g., fast, slow, or normal) fall into this category. **As detailed in our previous rebuttal, we contend that only specific paralinguistic attributes, accent, necessitate significant time and meticulous effort from human experts for rigorous filtering.** Furthermore, each expert was tasked with strictly filtering for **implicit dialogue samples** that received a score not equal to 1, averaging only 400 samples per expert. **We consider this workload of 400 samples per expert to be reasonable** and consistent with the standard operating procedures of industrial-scale data teams. The ChatReward-30K dataset is planned for open-source release. Thank you again for your constructive engagement.
>
> Best wishes.

---

### Official Review · Reviewer_mJjQ · 2025-10-29

**Soundness:** 2
**Presentation:** 2
**Contribution:** 3
**Rating:** 4
**Confidence:** 4

**Summary:**

This paper proposes WavReward, a novel model designed to evaluate the performance of spoken dialogue systems that can both understand and generate speech (e.g., GPT-4o-audio, Qwen2.5-Omni, Moshi). Existing evaluation methods mainly focus on text, ignoring non-verbal cues such as tone, emotion, and rhythm. As a result, they cannot accurately measure the true conversational and emotional quality of modern end-to-end speech models. WavReward fill this gap. Extensive experiments show the superior performance of WavReward.

**Strengths:**

1. It is an interesting and practical idea to train a reward model that evaluates generated speech based on the dialogue context.
2. The model outperforms existing baselines and SFT results. Experiments on RealDialogue demonstrate that the trained model not only fits well to the benchmark but also generalizes effectively to real-world conversations.

**Weaknesses:**

I think the paper is well-motivated and shows promising potential.

1. Since the thinking process lacks explicit ground truth and is learned through reinforcement learning, it is crucial to examine whether the reasoning process is meaningful and whether it provides useful insights into how the generated speech aligns with dialogue context. Adding human evaluation results to validate this aspect would greatly strengthen the paper.
2. There is a large body of previous work (e.g., [1][2]) that employs Speech Language Models (SLMs) as speech quality evaluators and reasoners. Please include a discussion of these related studies for completeness.
3. Please disclose the source and background of human annotators to ensure transparency and reproducibility.
4. It would be helpful to include more analysis of the proposed dataset, such as inter-annotator agreement. In my own experiments, the variance of MOS scores tends to be high. I am curious whether the variance is even larger in this more complex setting where annotators must consider dialogue context.
5. Could you provide more qualitative examples of the implied chat scenarios? This part seems highly subjective, and there may not be a single “gold” answer for many cases.

[1] Wang, Siyin, et al. "Qualispeech: A speech quality assessment dataset with natural language reasoning and descriptions."

[2] Chen, Chen, et al. "Audio large language models can be descriptive speech quality evaluators."

**Questions:**

### Suggestions
1. When a citation is used as a noun in a sentence, consider using $\citet$ instead (line 286).

### Questions
1. Is there any example where $s_{1}$ is low? Does it refer to the semantic aspect of the speech?
2. In lines 376–377, how did you determine which samples in RealDialogue are labeled as positive and which as negative?

**Details Of Ethics Concerns:**

Please disclose the source and background of human annotators to ensure transparency and reproducibility.

---

> ### Author Response · Authors · 2025-11-24
>
> We sincerely appreciate your generally positive assessment! We will address your concerns and questions below.
>
> ### **W1: Chain-of-Thought**
> Thank you very much for your expert suggestion. We have supplemented our paper with further experiments. Specifically, we leveraged the advanced Gemini-2.5-Pro model to evaluate the alignment between the input speech dialogue data and the corresponding CoT text, providing a binary classification of Good or Poor quality. The strict requirements imposed by the prompt included the CoT's semantic completeness, its correct analysis from both semantic and acoustic dimensions, and whether it faithfully reflects the genuine reasoning process of the dialogue. We tested the inference CoT text generated by WavReward on the ChatReward-30K-test dataset, with the results shown below:
>
> | Models | COT Text Acc↑ |
> |--------|--------|
> | WavReward | 96.6 |
>
> Furthermore, we randomly selected 100 samples from the inferred CoT text above and had human experts judge the quality of the CoT. The accuracy reached $97\%$. We have included these findings and the resulting conclusion in Appendix I in the revised manuscript .
>
> ### **W2: Related Work**
> We are grateful for your professional advice. We have expanded the relevant discussion in the Related Work section of the main text.
>
> It is worth noting that some recent work has attempted to use audio language models as evaluators. For instance, QualiSpeech[1] proposed a dataset for speech quality scoring (including the scoring process) and fine-tuned SALMONN using this data. However, WavReward's focus is on the dialogue scenario (semantic and acoustic) for spoken dialogue models, rather than mere speech quality assessment. Furthermore, WavReward is not a simple Supervised Fine-Tuning (SFT) approach; its overall framework is based on a more efficient Reinforcement Learning (RL) method, and it introduces the deep reasoning process (constrained by a separate think component), the nonlinear reward mechanism, and the multi-sample feedback mechanism. Additionally, recent work[2] has also attempted to use audio language models as MOS evaluators, proposing the ALLD method (which aligns the generated sequence of the audio LLM to a text LLM response and uses DPO fine-tuning). WavReward's training process differs significantly from the proposed ALLD method. Moreover, WavReward emphasizes performance in dialogue scenarios (both semantic and acoustic tasks), focuses on out-of-domain generalization, and is committed to open-sourcing the relevant code and data to advance the community.
>
> ### **W3: Human Experts**
> We utilized our company's professional LLM data annotation team, who possess profound experience with foundation language models. Due to company policies, we can only disclose this information and regret that specific names cannot be revealed.
>
> ### **W4: Human Preference Fluctuation**
> Thank you very much for your professional suggestion. In our practical experience, we found that different human annotators exhibit variability in their subjective preferences, **making the creation of scores from scratch much harder than verifying pre-assigned scores**. Therefore, the initial scores for ChatReward-30K were generated by a deterministic, automated rule set **firstly**(1 point for text error, 3 points for correct text but incorrect paralinguistics, and 5 points for correct text and correct paralinguistics). We use specific prompts to instruct GPT4 to automatically generate responses that are either textually correct (3-point/5-point) or textually incorrect (1-point), and then use rules to assign the correct paralinguistic label for the 5-point samples and the incorrect paralinguistic label for the 3-point samples. **Human experts are only required to verify the accuracy of the GPT generated text content and its conformity to the specified paralinguistic scenario (the model is accurate in most of cases)**, thereby easily guaranteeing scoring consistency.
>
> ### **W5: Examples**
> We have included specific examples of implicit dialogue scenarios in Appendix A of the manuscript.
>
>
> ### **All Questions**
> Thank you for your suggestion regarding the writing. We have made the revisions. We have provided an example of a 1-point (s1) response in Appendix A, which refers to a text-based error. Since the RealDialogue set contains only 120 samples, we had human experts manually score them and provide justification for the scores.
>
> **Thanks again for your professional suggestions! We are very willing to further discuss any new questions you may have and eagerly look forward to your continued support!**
>
> [1] Wang, Siyin, et al. "Qualispeech: A speech quality assessment dataset with natural language reasoning and descriptions."
>
> [2] Chen, Chen, et al. "Audio large language models can be descriptive speech quality evaluators."

---

> > ### Author Response · Authors · 2025-11-26
> >
> > Dear Reviewer mJjQ:
> >
> > We sincerely appreciate your generally positive assessment! As we are in the discussion period, we would be grateful if we could hear your feedback regarding our answers to the reviews. We would be happy to answer and discuss if you have further comments. Thanks for your expertise and precious time!
> >
> > Best regards.
> >
> > The WavReward Authors.

---

### Official Review · Reviewer_fsHd · 2025-10-30

**Soundness:** 3
**Presentation:** 3
**Contribution:** 4
**Rating:** 8
**Confidence:** 5

**Summary:**

This paper focuses on the core challenges in evaluating end-to-end speech dialogue models. Traditional evaluation methods heavily rely on text transcripts, failing to capture and measure the rich paralinguistic information inherent in speech interactions. This limitation hinders comprehensive assessments of a model's overall capabilities. To address this, the authors propose WavReward—a universal reward evaluator based on audio language models—aimed at providing a unified and in-depth assessment of both a model's IQ and EQ.

WavReward's core architecture builds upon audio language models and is optimized through reinforcement learning, enabling direct processing of end-to-end speech dialogue data. To enhance evaluation accuracy and interpretability, the framework integrates CoT, prompting models to generate justification before assigning scores. Additionally, a novel nonlinear reward mechanism imposes exponential penalties on results exhibiting significant scoring deviations. Furthermore, the training process employs a positive-negative multi-sample feedback strategy. By comparing multiple superior and inferior responses to the same question, this approach significantly enhances the evaluation model's discriminative capabilities.

To support model training and validation, this work constructs ChatReward-30K, a large-scale, high-quality speech dialogue preference dataset covering multiple dimensions. WavReward significantly outperforms all baseline models in objective accuracy. More importantly, in real-world subjective A/B tests, its evaluation results demonstrate high consistency with human preferences, proving the method's effectiveness and practical value.

**Strengths:**

- How to scientifically, comprehensively, and efficiently evaluate speech dialogue models is a critical issue that demands urgent attention. This paper precisely addresses this pain point, making its research highly timely and significant.
- The ChatReward-30K dataset represents a significant contribution. The authors provide a detailed account of its construction process, covering multiple dimensions including content, acoustics, and implicit/explicit aspects. This fills a gap in the field by offering a high-quality, comprehensively annotated evaluation benchmark.
- The experiment compared multiple state-of-the-art audio-language models (including direct inference and supervised fine-tuning), making the results more compelling. Evaluation was conducted using both in-domain (ChatReward-30K-test) and out-of-domain (RealDialogue) data, demonstrating the model's generalization capability. This work combines objective accuracy metrics with subjective A/B testing to validate WavReward's superiority from multiple perspectives. The high consistency between results and human judgments indicates alignment between the model and human preferences.

**Weaknesses:**

- The ChatReward-30K dataset is primarily synthesized using TTS models. This process may introduce synthetic biases or artifacts (such as unnatural prosody), raising questions about its ability to generalize to real human conversations.
- The generated data is filtered by WER and SER. However, no systematic filtering was conducted for accents, pitch , and other factors. Consequently, samples with poor synthesis quality—such as those generated by TTS models that deviate from the specified accent in the instructions—may adversely affect both training and testing.

**Questions:**

- Could you elaborate on the specific implementation of the “think format reward” (Rf)? During training, how does the system automatically determine whether the CoT generated by the model is compliant, thereby awarding either 5 points or 0 points? Is this based on keyword matching or an inspection of the output structure?
- Is there a text-based LLM serving as an upper-bound baseline with the input of dialogue transcriptions (with paralinguistic information provided as textual descriptions)? It could help compare WavReward's theoretical upper limit.
- In Table 1, most baselines exhibit similar performance on ChatReward and RealDialogue, while WavReward significantly outperforms on ChatReward compared to RealDialogue. Does this indicate that WavReward risks overfitting on the in-domain test set, or could other factors be responsible?

---

> ### Author Response · Authors · 2025-11-24
> **Rebuttal to Reviewer fsHd**
>
> We are very grateful for your initial high score and recognition of WavReward! Please allow us to provide more detailed responses to your questions.
>
> ### **W1: Generalize to real human conversations**
>
> Thank you for your suggestion. Evaluating generalization ability is indeed a crucial experiment for WavReward. Therefore, we tested WavReward's performance on the RealDialogue test set, which consists of genuine Human-Machine dialogue. In this set, the query part is recorded by real human speakers, and the answer part is generated by authentic dialogue models (Llama-Omni and Kimi-Audio). This dataset is out-of-domain compared to the TTS-synthesized ChatReward-30K used for training. We found that WavReward exhibits excellent generalization performance, and we plan to open-source the WavReward model.
>
> ### **W2: TTS pipline**
> We appreciate your feedback. It is true that we did not employ an automated filtering policy for some specific acoustic features, such as accents. We primarily ensure the quality of the TTS pipeline through two main approaches. Firstly, we utilized the most powerful TTS models available at the time for different paralinguistic acoustic information (GPT-4o-mini-TTS, Step-Audio-TTS, and CosyVoice2). We observed that GPT-4o-mini-TTS excels at synthesizing specified accents (e.g., Indian English), while Step-Audio-TTS performs relatively better at synthesizing specified ages (e.g., child voices). Secondly, we deployed human experts to manually filter out audio samples where the TTS-synthesized paralinguistic effects were of poor quality. it is precisely one of the reasons we believe ChatReward-30K is a valuable contribution to the community.
>
> ### **Q1: Think format reward**
> In WavReward, the model provides a corresponding reward based on whether it detects the output of the special token <think></think>.
>
>
> ### **Q2: Theoretical Upper Bound of WavReward**
>
> Given the substantial gap between evaluating dialogue paralinguistics by first converting them to text versus direct end-to-end evaluation of the spoken dialogue (e.g., accuracy of extracting paralinguistic text from speech, whether all paralinguistic acoustics need to be transcribed, and how to extract text for unseen paralinguistic acoustics outside the defined rules), we did not test the upper bound of a purely text-based foundation model as a reward model. In our subsequent experiments, we found that the capability ceiling of the reward model is partially determined by the base audio language model. If we replace Qwen-Omni with the latest Gemini, WavReward's performance is further improved, allowing for high-quality scoring across a wider range of scenarios.
>
>
> ### **Q3: Analysis of Stability**
> Thank you for your suggestion. We have further compiled the average results for the experiments in Table 1, which are shown below. We find that: (1) The performance of some models, such as SALMONN and GPT-4o-Audio, also shows some fluctuation between the ChatReward-30K and RealDialogue test sets. Other models, such as Qwen2.5-Omni, exhibit less fluctuation. This indicates that the fluctuation observed in the WavReward model falls within a reasonable range. (2) WavReward indeed performs better on the in-domain test set (ChatReward-30K-test) compared to the out-of-domain RealDialogue test set. However, the performance improvement of WavReward over Qwen2.5-Omni on the RealDialogue set confirms that WavReward possesses strong generalization performance (i.e., it is not overfitted).
> | Models | ChatReward-30K-Test(average) ↑ | RealDialogue ↑ |
> |--------|--------|--------|
> |  SALMONN | 30.6 |  19.2 |
> | GPT-4o-audio| 78.9 | 57.6 |
> | Qwen2.5-Omni| 53.5 | 51.7 |
> | WavReward | 92.5 | 80.8|
>
>
> **Thank you very much for your professional suggestions!**

---

> ### Comment · Reviewer_fsHd · 2025-11-27
>
> I appreciate the authors' detailed response, specifically regarding the filtering strategy for acoustic features (W2) and the stability analysis (Q3).
>
> While the rebuttal addresses my questions, the authors' confirmation that no automated filtering was applied for fine-grained acoustic attributes (like accents) and the performance gap observed between in-domain and out-of-domain settings have led me to reconsider the robustness of the current implementation. Considering these specific limitations in data quality control and generalization, I believe a score of 6 is a more accurate reflection of the paper's current state. I still consider this a solid contribution, but these constraints are notable.

---

> > ### Author Response · Authors · 2025-11-27
> > **Further Discussion with  Reviewer fsHd**
> >
> > Thank you very much for your reply! and in particular for ultimately assigning a positive score of 6.
> >
> > Regarding your concerns, we maintain the following positions:
> >
> > 1. On High-Level Acoustic Information (e.g., Accent) in TTS: When developing WavReward, we found that we did not possess a fully satisfactory automated tool capable of determining the acoustic accuracy of accents (specifically Indian, Singaporean, Canadian, or Australian English accents) during the evaluation process. If you can recommend an automated accent detection tool available before the ICLR submission deadline (i.e., before June 2025), we would be very willing to supplement our paper with the relevant experiments! Consequently, for the acoustic information concerning accents, we relied on expert subjective judgment and filtering.
> >
> > 2. On Performance Drops with Baselines and WavReward on the RealDialogue Test Set: As we introduced on Lines 396–399 of our paper: "In the RealDialogue dataset, we observed that certain dialogues have extended durations, and there are instances of poor audio quality, such as distorted electronic sounds. These factors present a more rigorous challenge for evaluating the model’s performance in unseen, real-world scenarios." Therefore, we believe that RealDialogue is inherently more challenging, and the performance and quality of both the ChatReward-30K dataset and the WavReward model are within our expectations for this challenging dataset.
> >
> > Thank you once again for your willingness to assign a score of 6. WavReward is also indeed one of our valued works.
> >
> > Best Wishes.
> >
> > The WavReward Authors.

---

### Official Review · Reviewer_mqd2 · 2025-11-01

**Soundness:** 3
**Presentation:** 3
**Contribution:** 3
**Rating:** 4
**Confidence:** 2

**Summary:**

This paper introduces WavReward, a novel reward model designed to evaluate end-to-end spoken dialogue models by directly processing speech-to-speech dialogues. The authors correctly identify a significant gap in the evaluation of such models, as existing benchmarks rely on transcribing audio to text, thereby losing crucial paralinguistic information (e.g., emotion, tone, accent). WavReward is an audio language model trained via reinforcement learning, incorporating three key innovations: a Chain-of-Thought (CoT) reasoning process, a nonlinear reward function, and a multi-sample feedback mechanism. To support this, they also introduce ChatReward-30K, a comprehensive dataset for training and evaluating audio reward models, covering both explicit and implicit dialogue scenarios with human-expert scores. Experiments show that WavReward significantly outperforms strong baseline models, including GPT-4o-audio, in both in-domain and out-of-domain settings.

**Strengths:**

1. Well-Motivated Problem: The paper effectively highlights a critical, underexplored problem: the lack of evaluation methods that can assess the full spectrum of spoken dialogue models' capabilities, particularly the paralinguistic and emotional quotient (EQ), without converting speech to text. The motivation is clear and strong.

2. Comprehensive Solution (Model & Dataset): The work is holistic, proposing both a novel model (WavReward) and a supporting dataset (ChatReward-30K). This two-pronged approach is a significant contribution to the community.

3. Methodological Components: The integration of CoT reasoning, a nonlinear reward function, and multi-sample feedback is a sophisticated and well-justified approach. The ablation studies (Table 1) effectively validate the importance of each component.

4. Extensive and Convincing Evaluation:
The evaluation is thorough, testing on a wide range of acoustic attributes (age, accent, emotion, etc.) and dialogue types (content, explicit, implicit).
The use of a real-world, out-of-domain dataset (RealDialogue) is a strong point, demonstrating the model's robustness and practical applicability.
Including both objective metrics (accuracy) and subjective A/B tests with human experts greatly strengthens the validity of the claims. The high win rates (77-83%) against powerful baselines like GPT-4o-audio are impressive.

5. Strong Empirical Results: The performance gains over state-of-the-art baselines are substantial and clearly demonstrated. The leap from Qwen2.5-Omni's 53.4% to WavReward's 91.5% on objective accuracy is a powerful result.

**Weaknesses:**

1. Dataset Construction Details:
While the dataset construction process is described in stages, more details would be beneficial. For example, how were the "human experts" selected and calibrated to ensure scoring consistency?
The prompt templates in the appendix are helpful, but a more detailed discussion of the challenges in generating high-quality, diverse implicit dialogues would be valuable.

2. Computational Cost and Scalability:
The computational cost of WavReward's training and inference is not discussed. The multi-sample sampling and CoT reasoning likely introduce significant overhead compared to direct inference. A discussion of this trade-off between evaluation quality and cost is important for practical adoption.

3. Clarity of Writing and Terminology:
The writing can be improved for clarity. Some sentences are long and complex, making them difficult to parse (e.g., the first sentence of the abstract).
The term "positive-negative multi-sample sampling mechanism" is used in the abstract and text, but the method section describes it as selecting samples at different quality levels. The terminology should be made consistent.
The acronym "GT" is used before being defined.

**Questions:**

1. Baseline Fine-tuning: For the supervised fine-tuning baselines (Qwen2.5-Omni w/ Full-param & LoRA), what was the exact training objective? Was it a regression loss (MSE) between the predicted and ground truth scores?

2. Generalization: How do you envision WavReward generalizing to entirely new paralinguistic features or languages not present in ChatReward-30K? Is the model learning a general evaluation principle, or is it heavily tuned to the specific attributes in your dataset?

3. Inference Speed: What is the relative inference latency of WavReward compared to a baseline like Qwen2.5-Omni with direct inference? This is a practical concern for using such an evaluator in development pipelines.

---

> ### Author Response · Authors · 2025-11-24
> **Rebuttal to Reviewer mqd2(1/2)**
>
> We sincerely appreciate your highly positive assessment of WavReward in the strengths section of your review. Please allow us to provide a more detailed and specific response to the questions and concerns you raised.
>
> ### **W1: Dataset Construction Details**
>
> We are very grateful for your insightful suggestion. In our practical experience, we found that different human annotators exhibit variability in their subjective preferences and **creating scores from dialogue is substantially more challenging than verifying pre-assigned scores**. Therefore, the scores for the ChatReward-30K dataset were generated by a deterministic, automated rule set **firstly**. The scoring logic is as follows: (1) 1 point for text errors, (2) 3 points for correct text but incorrect paralinguistics (e.g., tone), and (3) 5 points for correct text and correct paralinguistics. To implement this, We use specific prompts to instruct GPT-4 to automatically generate responses that are either textually correct (3-point/5-point) or textually incorrect (1-point). For explicitly instructed dialogue scenarios (e.g., speaking in a happy tone), we then use rule-based assignment to designate the correct paralinguistic label for the 5-point samples and the incorrect paralinguistic label for the 3-point samples. Human experts are tasked only with verifying the accuracy of the GPT-generated text content and its conformity to the specified paralinguistic scenario. Since the GPT4 achieves high fidelity (accurate in $\approx 99\%$ of cases) instruction following, we can easily guarantee scoring consistency. Regarding the selection of human experts, this task was handled by our company's existing, high-quality team of LLM data experts.
>
> Regarding the generation of high-quality implicit dialogue data. To ensure textual diversity, we designed various distinct dialogue scenarios (as detailed in Line 357 and Line 874 of the revised manuscript). To ensure textual accuracy, we incorporated relevant constraints and example parameters into the prompts (as detailed in Line 876 and Line 880). Furthermore, we observed that implicit dialogue scenarios are often more distinctly correlated with emotion, speaking rate, and volume. Thus, we have augmented the data in these dimensions for the implicit dialogue contexts. When constructing the dialogue dataset with distinctly different scores (3 points vs. 5 points), we specified significantly disparate emotions (e.g., Happy vs. Angry), rather than subtle differences (e.g., Happy vs. Gentle), to ensure a clear distinction between the reward signals. We have included these details in Appendix D.
>
> ### **W2: Computational Cost and Scalability**
>
> We sincerely appreciate your valuable suggestion.The multi-sample sampling technique is introduced during the training phase specifically. Its purpose is to enable the model to better discriminate between the quality of different responses generated for the same query, thereby refining its scoring capability. Critically, during inference, WavReward generates only a single score. **Consequently, multi-sample sampling introduces no additional latency at the inference stage.**
>
> Regarding the latency associated with the Chain-of-Thought mechanism, we conducted further evaluation using the RealDialogue test set on a single H20 96G GPU. We measured the Real-Time Factor (RTF) for both the baseline (WavReward w/o cot think) and the full model (WavReward). RTF is defined as the time taken to generate the score divided by the duration of the input dialogue data. The results shown below confirm that while CoT provides a substantial performance gain (as demonstrated in Table 1 of our paper), it does introduce a certain degree of increased inference latency. This aligns with the predominant expectation.
> | Models | RTF ↓ |
> |--------|--------|
> | WavReward w/o cot think | 0.172 |
> | WavReward | 0.318 |
>
>
>
> In addition to latency, we have also thoroughly analyzed the quality of the generated CoT text. Specifically, we employed the advanced Gemini-2.5-Pro model to evaluate the alignment between the input speech dialogue data and the corresponding CoT text, classifying the relationship as either Good or Poor (a binary classification). The evaluation prompt strictly enforced the following requirements: semantic completeness of the CoT, correct analysis along both semantic and acoustic dimensions. accurate reflection of the true reasoning process behind the dialogue. We tested this on the CoT text generated by WavReward on the ChatReward-30K-test dataset. The results are as follows: These results also confirm the quality and fidelity of our CoT mechanism.
>
> | Models | COT Text Acc↑ |
> |--------|--------|
> | WavReward | 96.6 |
>
> These results have been supplemented in Appendix H of the revised manuscript.
>
> ### **W3: Writing**
> Thanks for your suggestions concerning the manuscript's writing. We have incorporated the necessary revisions throughout the paper.

---

> ### Author Response · Authors · 2025-11-24
> **Rebuttal to Reviewer mqd2(2/2)**
>
> ### **Q1: Baseline details**
> the exact training objective is LLM loss (Cross-Entropy Loss) between the predicted and ground truth scores.
>
> ### **Q2: Generalization ability**
> We greatly appreciate your suggestion. The generalization ability experiment is indeed one of the key highlights of WavReward. For this reason, we specifically constructed the RealDialogue test set(a completely out-of-domain, real human-to-machine dialogue dataset). The answers in RealDialogue source from the dialogue models Llama-Omni and Kimi-Audio which differs significantly from the training set (ChatReward-30K) in terms of content, duration, and audio quality. As shown in Table 1, our results on this out-of-domain set demonstrate the superior generalization performance of WavReward. We are committed to making the full code and data publicly available if paper is accepted. Regarding language support, WavReward currently supports English and Chinese only.
>
> ### **Q3: Inference Speed With Qwen2.5-Omni-7B**
>
> When compared against the Qwen2.5-Omni-7B(Thinker) baseline, WavReward does not introduce any additional inference latency.
>
> **Once again, thanks for your professional and insightful suggestions! We are eager to further discuss any new questions you may have and look forward to your continued support!**

---

> > ### Author Response · Authors · 2025-11-26
> >
> > Dear Reviewer mqd2:
> >
> > We sincerely appreciate your highly positive assessment of WavReward in the strengths section of your review. As we are in the discussion period, we would be grateful if we could hear your feedback regarding our answers to the reviews. We would be happy to answer and discuss if you have further comments. Thanks for your expertise and precious time!
> >
> > Best regards.
> >
> > The WavReward Authors.

---

### Author Response · Authors · 2025-11-27

Dear Reviewers:

As we are in the discussion period, we would be grateful if we could hear your feedback regarding our answers to the reviews. We would be happy to answer and discuss if you have further comments. Thanks for your expertise and precious time.

Best regards.

---

### Author Response · Authors · 2025-12-02
**Rebuttal Summary From Authors**

Dear ACs:

We are immensely grateful for all the reviewers' valuable suggestions and constructive feedback. While the initial aggregate score for WavReward was 2448 (average 4.5), we wish to emphasize that at the initial stage, all low-score reviewers provided substantial positive comments on the paper. **Specifically: Reviewer mqd2 (Initial score: 4) recognized several strengths about "Well-Motivated Problem, Comprehensive Solution (Model & Dataset), Methodological Components, Extensive and Convincing Evaluation, and Strong Empirical Results" in the Strengths section. Reviewer mJjQ (Initial score: 4) noted in the Weaknesses section that WavReward "is well-motivated and shows promising potential". Reviewer 8qWV (Initial score: 2) expressed a positive attitude and willingness to increase his score during the rebuttal period.** Due to unforeseen circumstances related to the ICLR conference schedule, the two reviewers who initially scored 4 were unable to reply in time. However, We sincerely believe that our revised submission has effectively addressed the issues they raised.
During the rebuttal, we provided detailed supplementary information regarding technical specifics such as GRPO, training procedures, dataset examples, and general writing clarity improvements. Below, we further summarize and respond to the most critical issues raised:

### **1. Computational Cost in WavReward [Reviewer mqd2]**
The inference efficiency of WavReward without CoT (Chain-of-Thought) thinking is identical to that of Qwen2.5-Omni. The integration of the CoT reasoning paradigm, while leading to a performance gain, introduces an expected marginal increase in latency (from $0.172$ to $0.318$). We consider this result to be satisfactory and well within expectations.

### **2. Quality of CoT Rationales [Reviewer mJjQ]**
The quality of WavReward's CoT rationales was rigorously assessed using Gemini-2.5-Pro (evaluating semantic consistency, comprehensiveness, and rationality), yielding a satisfactory agreement score of $96.6\%$. This evaluation was further corroborated by manual verification from human experts, who found their preference to be highly consistent with the Gemini-2.5-Pro.

### **3. Gap between ChatReward-30K and RealDialogue, and Overfitting Concerns [Reviewer fsHd]**
We acknowledge the slight performance decrease of WavReward and baselines on the RealDialogue test set compared to ChatReward-30K. This outcome is expected, as RealDialogue is a highly robust, authentic, and in-the-wild dataset containing real person-dialogue model interactions, featuring extended durations, poor audio quality (e.g., distorted electronic sounds) and so on. Therefore, we maintain that both the ChatReward-30K dataset and the WavReward model demonstrate commendable and expected performance given these challenging conditions.

### **4. Data Pipeline of ChatReward-30K [Reviewer fsHd]**
Given that the ChatReward-30K dataset incorporates a wide array of paralinguistic information, we employed automated filtering primarily for WER and emotion accuracy. For highly nuanced paralinguistic attributes, specifically accents (e.g., Canadian English, Singaporean English, Indian English), **there were no highly effective, commercially available accent classification models before the deadline of ICLR**. Consequently, we did rely on subjective filtering by human experts for accent.

### **5. Human Expert Preference and Quantity [Reviewer 8qWV]**
The five experts involved in the ChatReward-30K creation are members of the professional LLM data team in our company. In our detailed response to Reviewer 8qWV, **we fully elaborated on the process used by the experts in curating the ChatReward-30K dataset**. Pertaining to the paralinguistic mentioned by Reviewer 8qWV, we further underlined that the per-expert workload of 400 samples for implicit dialogue scenarios and filter the accent paralinguistic features is appropriate. A demo and detailed description of the ChatReward-30K dataset have been included in the Appendix, and both the ChatReward-30K and the WavReward will be open-sourced upon acceptance.

In summary, we introduce WavReward, the first reward model designed specifically for end-to-end spoken dialogue systems, successfully demonstrating that audio language models are effective evaluators. Its enhanced capability stems from a reasoning-based assessment process and novel mechanisms (nonlinear reward feedback and diverse sampling) during reinforcement learning. Rigorous ablation studies and main results confirm its strong performance, even on the out-of-domain RealDialogue test set. Both the WavReward model and the ChatReward-30K dataset will be open-sourced to benefit the community.

**Thank you once again for the ACs' professional dedication and invaluable time spent in evaluating our work！**

Best wishes.

The WavReward Authors.

---

### Meta-Review · Area_Chair_agh3 · 2026-01-10

**Summary:**

The paper proposes WavReward, an audio-LLM-based reward model for evaluating spoken dialogue systems, with a dataset ChatReward-30K. While reviewers appreciated the strong motivation and the novelty of addressing the evaluation gap for paralinguistic features (EQ/IQ), the consensus leans towards rejection. The primary concerns stem from the validity and robustness of the ChatReward-30K dataset, which serves as the foundation for the proposed model. Some critical issues include the reliance on a very small pool of human experts (5 individuals) to verify a large-scale dataset, the use of synthetic data generation (TTS) without sufficient automated quality control for fine-grained attributes (e.g., accents), and signs of overfitting to the synthetic training distribution compared to real-world data.

**Reviewer Concerns:**

**Concerns addressed by rebuttal:**
- The authors provided Real-Time Factor (RTF) analysis showing that CoT introduces latency but remains within reasonable bounds (addressed mqd2).
- An evaluation using Gemini-2.5-Pro and human experts was conducted to verify the semantic and acoustic alignment of the generated rationales (addressed mJjQ).
- The authors clarified the use of GRPO (Group Relative Policy Optimization) and corrected the Thinker model references and naming conventions (addressed 8qWV, mqd2).
- Baselines and implementation details (e.g., SFT objectives) were clarified (addressed mqd2, 8qWV).

**Still outstanding concerns:**
- The reviewer explicitly retained reservations about the use of only 5 human experts to manually verify the ChatReward-30K dataset. The concern is that this small number is insufficient to capture the diversity of human judgment on subjective paralinguistic aspects (e.g., tone, emotion) for a "generalist" benchmark, unlike objective tasks. The authors' defense regarding "efficiency" did not resolve the fundamental issue of potential bias and lack of distributional representativeness.
- The reviewer fsHd downgraded their score (from 8 to 6) after the rebuttal. The acknowledgement that no automated filtering was applied for fine-grained acoustic attributes (like accents) and that the method relies heavily on manual filtering by the small expert pool raised significant robustness concerns. Furthermore, the performance gap between the in-domain test set (ChatReward-30K) and the out-of-domain set (RealDialogue) suggested potential overfitting to the synthetic generation pipeline.
- While the authors stated the annotators were an "internal team," the lack of detailed demographic or background information (beyond them being "experts") and the absence of rigorous inter-annotator agreement analysis for the *subjective* components (beyond rule-based verification) remain a weakness for a benchmark paper (Reviewer mJjQ).
- The model shows a significant performance drop when moving from synthetic test sets to the real-world RealDialogue dataset. Given that the RealDialogue set is very small (only 120 samples), the claim that the model generalizes well to real human speech is not statistically robust. (Reviewer 8qWV, fsHd)

**Reviewer Scores:**

- Reviewer mqd2 (4): Likely to remain 4. The reviewer's broader concern about Dataset construction details was only superficially addressed by stating the experts were an "internal team." Given that the core weakness of the paper (dataset validity) remains unresolved, the reviewer would have found little justification to raise the score above the acceptance threshold.
- Reviewer fsHd (6): Score changed from 8 to 6. The reviewer stated the limitations in data quality control and generalization led to a lowered assessment.
- Reviewer mJjQ (4): Likely to remain 4. The reviewer found the work promising but the rebuttal regarding the dataset's subjective validation was likely insufficient to warrant a significant score increase.
- Reviewer 8qWV (4): Score changed from 2 to 4. The reviewer acknowledged the clarifications but explicitly stated that the reservation regarding the "5 experts" issue prevented a higher score.

---

### Decision · Program_Chairs · 2026-01-26

Reject